# Long-Term Space Nutrition: A Scoping Review

**DOI:** 10.3390/nu14010194

**Published:** 2021-12-31

**Authors:** Hong Tang, Hope Hui Rising, Manoranjan Majji, Robert D. Brown

**Affiliations:** 1College of Landscape and Tourism, Gansu Agricultural University, Lanzhou 730070, China; tangh@gsau.edu.cn; 2Department of Landscape Architecture and Urban Planning, Texas A&M University, College Station, TX 77843, USA; rbrown@arch.tamu.edu; 3Department of Aerospace Engineering, Texas A&M University, College Station, TX 77843, USA; mmajji@tamu.edu

**Keywords:** long-term space tasks, astronauts, dietary deficiencies, adverse living environment, nutritional strategies, fresh food, self-sufficient, microgravity, space nutrition, space food systems

## Abstract

This scoping review aimed to identify current evidence and gaps in the field of long-term space nutrition. Specifically, the review targeted critical nutritional needs during long-term manned missions in outer space in addition to the essential components of a sustainable space nutrition system for meeting these needs. The search phrase “space food and the survival of astronauts in long-term missions” was used to collect the initial 5432 articles from seven Chinese and seven English databases. From these articles, two independent reviewers screened titles and abstracts to identify 218 articles for full-text reviews based on three themes and 18 keyword combinations as eligibility criteria. The results suggest that it is possible to address short-term adverse environmental factors and nutritional deficiencies by adopting effective dietary measures, selecting the right types of foods and supplements, and engaging in specific sustainable food production and eating practices. However, to support self-sufficiency during long-term space exploration, the most optimal and sustainable space nutrition systems are likely to be supported primarily by fresh food production, natural unprocessed foods as diets, nutrient recycling of food scraps and cultivation systems, and the establishment of closed-loop biospheres or landscape-based space habitats as long-term life support systems.

## 1. Introduction

From time immemorial, fresh and packaged food onboard long-term transportation systems has been a topic of intense research [1,2]. In addition to facilitating Earth exploration, and most famously, the discovery of the new world, nutrition became a commercial entity and ushered in a new age in human exploration [3,4,5,6]. The Industrial Revolution and the ensuing progress of humanity can be largely attributed to the nutrition industry, which provided capacitance against natural disasters and calamities, while ensuring food security for the burgeoning world population. Space exploration in the mid-20th century has also greatly benefited from the advancing trends in the food industry. Looking into the future, a variety of new techniques and technologies continue to be researched to harness resources from the harsh space environment to provide a sustainable means of space nutrition, while influencing the food preservation and culinary practices of future generations [7,8].

### 1.1. Backgrounds

Humans have been involved in manned spaceflight for the past five decades, with the International Space Station (ISS) as the main destination for short-term missions. Major space agencies’ extensive plans for a long-term human presence in space have also motivated missions to the Moon and Mars [9,10,11]. Extensive research activities are being carried out by various researchers to support long-term human presence in space. One notable example is the 8-month isolation mission called the Hawaii Space Exploration Analog and Simulation (HI-SEAS) III expedition. This recent research work by the National Aeronautics and Space Administration (NASA) involves space exploration simulation and modeling in the HI-SEAS habitat, a dome of 135.8 square meters with food and life support systems designed to simulate the conditions on Mars. To investigate the best way for astronauts to maintain optimal nutritional intake during long-term life on Mars or the Moon, researchers at the HI-SEAS facility explore new forms of foods and food allocation strategies for deep-space manned missions so that functional foods can be formulated and processed for consumption by astronauts in outer space [12].

For manned short-duration missions, astronauts rely on nutritional supplies carried from the Earth and sustained by specialized delivery missions. These food products have a shorter shelf life than the duration of long-term space missions to Mars. Moreover, the techniques used for processing these food products makes them limited in terms of satisfying consumers’ nutritional requirements. Due to the tradeoffs between taste, nutrition, storage, and packaging considerations, the limited palatability of processed space foods can lead to menu fatigue, preventing astronauts from taking in enough food to acquire sufficient nutrients. On the other hand, processed space foods cannot provide the full range of diverse nutrients necessary to help astronauts effectively combat bone loss [13,14,15], muscle atrophy [15,16], cardiovascular dysfunction, upright intolerance, and other physiological challenges associated with the extreme environment in space [17,18,19]. While exercise and medical support are both necessary to overcome some of these challenges, special nutritional resources are vital for in-flight adaptation and post-flight recovery.

To compensate for the aforementioned deficiencies in the existing space nutrition system, which largely relies on processed food, fresh food materials are necessary. In fact, various digestive problems have been identified and are thought to be associated with the lack of fresh fruits and vegetables and fiber-rich meals, even with the use of supplements to increase the diversity of nutrients in space nutrition [12]. However, the recent space dietary research from EuroMoonMars IMA HI-SEAS II (EMMIHS-II) mission significantly limited food choices to freeze-dried foods and prohibited the use of fresh foods because fresh foods were considered too microbiologically fragile to meet the sanitary needs of the missions [12]. While freeze-dried foods make no contribution to psychological wellbeing, fresh foods provide a sense of familiarity to help sustain the mental health of astronauts and future space tourists [12]. As relying solely on packaged foods may not be feasible for long-term space exploration, advanced lighting, irrigation, and greenhouse systems have been developed to optimize plant growth in outer space to make fresh space foods possible [12]. Deep space exploration resupply programs will require self-sufficient closed-loop ecosystems of resource production and regeneration to provide renewable resources that minimize energy consumption [20,21,22] and maximize the use of higher plants and other autotrophs [23,24,25,26,27,28,29].

### 1.2. Rationale and Objectives

The most recent approach in space nutrition research has focused on adapting dietary plans to each astronaut while proposing alternatives to address food intolerances and aversions among a small group of astronauts [12]. However, this approach does not produce a generalizable evidence-based dietary plan for a much larger group of humans than the six astronauts that participated in the study. There is a growing demand to advance space nutrition systems in response to the emerging commercial space needs. The advent of companies such as SpaceX, Blue Origins, and Virgin Galactic has opened space for recreational opportunities for humans [1]. Optimizing long-term space nutrition for the likely multi-cultural space tourists and inhabitants of the near future is essential to the success of commercial space activities and space tourism.

This scoping review serves as a first step towards developing evidence-based space menus and production systems for long-term exploration and occupation of other planets. The review aimed to use a multi-national perspective to scope the extent to which space nutrition meets the diverse dietary needs for the short-term and long-term adaptation of human psychophysiology in the adverse space environment. The objective of the scoping review is to identify knowledge gaps and to synthesize evidence around nutrient deficiencies and dietary strategies to inform hypotheses to be tested in future experiments with a larger sample of more diverse participants that have a wider range of psychophysiological baselines.

## 2. Materials and Methods

This scoping review was conducted using the Preferred Reporting Items for Systematic Reviews and Meta-Analyses Extension for Scoping Reviews (PRISMA-ScR) checklist. Some of the checklist items were not used because they were designed for meta-analyses, which are out of the scope of this review. As a result, relevant checklist (CL) items were reported in the article with item numbers instead of a stand-alone checklist form with page numbers. The article was identified as a scoping review in the title for checklist item 1 (CL-1). The abstract provided the structured summary required for CL-2. Section 1.2 provides the rational and objectives for CL-3 and CL-4.

### 2.1. Protocol and Registration Information Sources

CL-4 requires the review protocol and registration to be included under methods. As the international prospective register of systematic reviews (PROSPERO) does not accept the registration of scoping reviews, a registration number for this scoping review is not available. Figure 1 illustrates the PRISMA review protocol used by two independent reviewers between November 2019 and December 2021.

### 2.2. Search Strategies and Eligibility Criteria

The preliminary search for the initial pool of review and non-review articles was conducted using “space food and the survival of astronauts in long-term missions” as the search phrase. The review articles from the selection were used to identify emerging themes and recurrent keywords as eligibility criteria required by CL-6. The first round of title and abstract screening was conducted using the following three themes as eligibility criteria: (1) acquiring fresh food materials in the long-term space mission, (2) maintaining a self-sufficient space survival mode, and (3) establishing human habitats. The theme-relevant articles were further filtered using the list of keywords in Table 1 to identify the final collection of articles for full-text reviews.

### 2.3. Information Sources

For CL-7 on information sources, the preliminary search identified review and non-review articles from 14 databases (Table 2). The dates of coverage span from 1964 to 2021. The most recent search was executed on September 5, 2021.

### 2.4. Search and Selection of Sources of Evidence

In total, 5432 results were obtained from the preliminary search. After removing duplicates, 4320 papers were left for the first round of title and abstract screening using the three themes to select 535 papers for another round of title and abstract screening. The list of keywords from Table 2 were then used as inclusion criteria to identify 218 articles for full-text screening. Figure 1 shows the screening process and results as a flowchart using the PRISMA template.

## 3. Results

### 3.1. Current State of Space Nutrition

#### 3.1.1. Key Components of Space Nutrition

Nutrition has many important functions in space travel, from providing enough nutrients and meeting the metabolic needs of a healthy body to enhancing an individual’s emotional well-being. Nutrition also plays a key role in offsetting many negative effects of space travel, such as radiation exposure, immune deficiency, oxidative stress, and bone and muscle loss [30]. Therefore, space nutrition must provide reasonable support for optimal physiological and psychological wellbeing in space in addition to accommodating diverse tastes, variety, and acceptability. Space nutrition should meet the daily human needs for protein, fat, and sugar, as well as inorganic elements, trace elements, fat-soluble vitamins, and various water-soluble vitamins. Space nutrition contains 16 essential nutrients: protein, calcium, iron, vitamin A, vitamin C, thiamine, riboflavin, vitamin b-12, folate, vitamin D, vitamin E, magnesium, potassium, zinc, fiber, and pantothenic acid [31]. In-flight nutrition requirements are set by the World Health Organization (WHO) according to the daily requirements of people on Earth. Therefore, a macronutrient composition with an average of 15% protein, 30% lipids, and 55% carbohydrates is recommended as a bare minimum [31,32].

#### 3.1.2. The Evolution of Space Nutrition

The evolution of space nutrition has been well documented by nutritional literature and NASA [33,34,35,36]. In the early 1960s, the research on space food systems focused largely on calorie-dense, nutritious, and palatable food without provisions for specific food storage on the spacecraft for short-duration missions. American and Russian astronauts lost weight from consuming primarily aluminum tubes packed with minced meat, jam, and other paste food, as well as bite-sized cubes with high-calorie mixtures of protein, fat, sugar, fruit, or nuts. Although these space foods met the nutritional, sensory, and microbiologic prerequisites in ground-based tests, astronauts experienced menu fatigue [35]. While dehydrated food and ice were developed in collaboration with the U.S. army’s Natick laboratory, rehydrating foods was difficult until hot water became available for the Apollo program (1968 to 1972) to improve the taste of space food [35].

The missions became longer from the mid-1960s to the early 1970s. Consequently, increased variety, improved quality, and longer-term storage shifted the focus of space nutrition towards packaged foods such as cans, food bars, and retort pouches [35]. Retorting enabled food storage at ambient temperature for a long period by thermally sterilizing the food [35]. Eating from open containers with utensils became possible for the first time as rehydrated foods were made more prominent in the space food systems by the abundance of by-product water from the increasing use of hydrogen and oxygen fuel cells to power American spacecraft [34]. The character and flavor of rehydrated food are closer to those in the common diet on the ground. While satisfying the taste of astronauts and increasing food choices [37], dehydration also helped reduce storage space and power needs by minimizing the need for food refrigeration [34,35].

The rapid development of food refrigeration and heating equipment on manned spacecraft facilitated the use of thermal stabilization bags, canned fruit, irradiated meat, and freeze-dried food in subsequent stages of space flight. Today, the types and varieties of space food are quite close to the nutritional choices available terrestrially. Astronauts aboard the ISS are able to eat fresh vegetables, fruits, and heated soup for most meals. The food supply during space flight must be safe, nutritious, convenient, and compact, while meeting the psychological and taste requirements of astronauts under weightlessness or artificial gravity.

#### 3.1.3. Space Food Categories

The main space food categories are canned food, dehydrated food, medium moisture food, natural food, refrigerated food, fresh food, irradiated food, and functional food [30]. The first five types of space food are relatively mature and widely used, while the last three kinds of space food are popular foods being actively developed to meet the emerging needs of commercial and recreational space flights and long-term space missions. The ISS recently started to test the viability of an on-demand nutrient production system composed of a desiccated yeast strain and edible growth substrate to produce ready-to-consume nutrients for long-duration missions [38]. While on-demand nutrient production system may become a new space food category once it has been validated to be safe and feasible, it is the least familiar type of food for astronauts. Despite its short shelf life, fresh food has been and will still be necessary for improving space food acceptability. The ISS provides astronauts with fresh food, mainly fruits and vegetables for direct consumption or vegetable salads [41,42]. Irradiated food refers to food sterilized by irradiation. Although this method processes food in small quantities most of the time, irradiation-based sterilization can be mass produced, especially by exploiting the harsh radiative atmosphere in outer space. Currently, irradiated food on the ISS mainly includes meat and bread. Korean scholars developed ready-to-eat consumables such as nutrition bars, noodles, and two kinds of traditional Korean food (kimchi and cinnamon beverage) by using high-dose gamma-ray radiation treatment [41,42]. Functional food alludes to special nutrients with supplemental health functions as space food additives to help astronauts better cope with the adverse effects of space living conditions through absorbing the restorative effects of the supplementation in a long-term manner. Anti-radiation functional food is an example of such approach [42,43].

#### 3.1.4. Space Food Menus

After decades of effort, space food has enjoyed a diversification of variety and taste. To avoid monotony, the diets for the American and Russian astronauts are generally based on a 4-to-6-day cycle, during which the food is different every day except for the drinks. Much of the Russian space food is canned. Lamb with vegetables, beef with barley, sturgeon, and chicken rice are the meal options that typically appear on the Russian menu. These options can be heated in the microwave. There are also many dehydrated foods, such as tvorog, macaroni, tomatoes, fried rice, and shrimp. The general diet of the American astronauts is divided into A, B, and C meals: an A meal has peaches, roast beef, scrambled eggs, pancakes, cocoa, orange drinks, vitamin pills, and coffee; the B meal consists of pork mix, turkey sausage, bread, bananas, almond crackers, and apple drink; the C meal is composed of shrimp, steak, risotto, broccoli, cocktail, pudding, grape juice, and ice cream. Chinese aerospace recipes are mainly made of traditional Chinese dishes, such as eight treasure rice, tangerine beef, beef in soy sauce, lotus seed porridge, green tea, ink fish balls, beef balls, and other Asian delicacies.

### 3.2. Limitations of Existing Space Nutrition

The following limitations have been the major drivers for innovations that contribute to the advancement in space nutrition.

#### 3.2.1. Dominance of Processed over Fresh Food

At present, astronauts are provided mainly with processed and packaged food. Fresh vegetables and fruits can only be enjoyed in the early stage of a space mission due to limited storage time and high cost. It is estimated that a 3-year Mars mission with a crew of six would have a total energy expenditure of 12 megajoules per person per day, regardless of water requirements, and would carry 22 tons of water-containing food on the spacecraft [44]. Even if the water is completely recycled and the food is partially dehydrated, the transport costs are estimated to be very high (20,000 euros per kilogram) [40,45]. The ISS prioritizes processed food with minimal weight and high nutrient density due to the significant cost associated with transporting fresh food.

Astronauts’ interest in health-promoting food, including fresh vegetables, is on the rise. There is no substitute for a healthy diet related to vegetable intake because fresh vegetables contain many health-promoting properties, including vitamins, minerals, dietary fiber, and secondary compounds [77]. The lack of adequate fresh vegetables and fruits is one of the most significant current challenges of space nutrition.

#### 3.2.2. No Quality Advantage for Resource-Intensive Refrigerated and Frozen Food

Fuel cells provide water to astronauts as a by-product from energy generation. The ISS recently started using solar cells to harvest energy from the sun. However, water and electricity remain extremely valuable and scarce resources due to the weight constraint of a space shuttle. These water, power, and weight limitations continue to make it challenging for space missions to accommodate freezers and refrigerators. Until these limitations have been addressed, there is no quality advantage to using refrigerated and frozen food [31].

#### 3.2.3. Space Food Supply Is Restricted by Limited Transportation and Storage Space

Mission resources, including power, size, mass, crew time, and waste disposal capacity, must be considered when developing space nutrition systems. Misuse of these resources will affect the success of the mission. While food and resource use may be contradictory, both are critical to the success of the mission [31]. Due to the high resupply cost, it is unrealistic to rely on transporting materials from the Earth to support long-term space missions and human settlements on other planets. There is a need to develop regenerative and self-sufficient water, food, and energy production systems.

#### 3.2.4. Long-Term Space Nutrition Requirements for Food Storage and Cooking Methods

A long-term space nutrition system must maintain sensory palatability, nutritional efficacy, and safety over a period of 3 to 5 years. NASA has aimed to develop nutrient-dense and environmentally sustainable food compatible with the cooking processes for microgravity [31]. However, it is challenging to use these same cooking processes to make space food last for more than 3 to 5 years without changing the food quality and nutritional value at the expense of human health. Even with the development of artificial gravity, which will make more food preparation and preservation methods feasible, the adoption of Earth-based systems for long-term space nutrition requires a drastic reduction in the external input of resources and output of wastes.

#### 3.2.5. Diet Menu Fatigue

Food acceptability can be affected by eating habits. Food and mealtimes can help promote solidarity among astronauts, resulting in important psychological and social benefits [31]. Food and mealtimes play a key role in reducing the stress and boredom from prolonged task execution, while delicious food provides pleasure for the eater. The appearance, taste, texture, and smell of food can have a significant psychological impact on astronauts [48]. Despite the great variety of food available today, it is still not enough for long-term space missions that last several years. However, the importance of the space food supply to long-term space missions cannot be understated. During the simulated manned landing on Mars experiment of MARS500 in Russia, many subjects showed “diet menu fatigue” and even became tired of their favorite food [49].

#### 3.2.6. Lack of Nutrients to Cope with Extreme Conditions of Space

In addition to being nutritious and safe, space food needs to function as a countermeasure to the negative effects of spaceflight by including nutrients that help the human body and mind adapt to weightlessness and the extreme conditions of space [49,50,51]. While the lack of gravity and circadian rhythm are well-known and widely studied aspects of spaceflight travel, there is a need for a more comprehensive nutritional study on other ancillary conditions, such as food taste alteration (due to the changes in atmospheric pressure) and the adaptations of human digestive, olfactory, and perception systems to long-term space habitation.

### 3.3. The Influence of Adverse Space Environment on Astronauts’ Diet and Health

The space environment is quite different from the terrestrial environment on Earth. Astronauts are faced with several unfavorable conditions for human survival: microgravity, radiation, confined space, motion sickness, and circadian rhythm changes, as well as a low-pressure atmosphere that is low in oxygen and high in carbon dioxide. Human spaceflight data show that space environments characterized by microgravity and 90 min light/dark cycles trigger countless adaptive responses from almost all physiological systems. These adaptive responses can lead to a loss of body mass, fluid transfer, electrolyte imbalance, dehydration, constipation, loss of potassium, loss of calcium, loss of red blood cell mass, intestinal microecological disorder, and space motion sickness [188]. The diet and health of astronauts can be negatively impacted by the adverse living conditions during manned flight [52].

#### 3.3.1. Less Energy Intake and Weight Loss

Preliminary results from terrestrial studies have shown that an increased protein/carbohydrate ratio is correlated with long-term weight maintenance after weight loss [78]. Such weight maintenance strategies have yet to be tested against the averse living conditions in space. It is challenging for astronauts to maintain their energy balance during long space flights [79,80]. This negative energy balance leads to weight loss [79,80,95,96,97]. Astronauts typically lost 2% to 5% of their pre-flight weight [53,99,100,101,102]. In many cases, more than 10% weight loss was observed even though there was plenty of food on board [45]. If sustained, this could result in a weight loss of 5 kg per month [45]. A mission to Mars could result in an initial weight loss of 15% or more, which could have serious health implications [45].

On Earth, a long-term negative energy balance can lead to compromised muscle performance, impaired cardiovascular function, increased muscle fatigue, increased susceptibility to infection, impaired wound healing, altered sleep, and decreased overall well-being [102]. Chronic energy deficiency can exacerbate some harmful physiological adaptations to the space environment, resulting in cardiovascular dysfunction, bone density, muscle mass and strength loss, impaired exercise ability, and immunodeficiency [45,54,55,102]. These physiological changes may jeopardize the health and performance of the crew, as well as the overall success of the mission [52].

Astronauts need to consume more food to offset the decrease in their energy intake due to microgravity, small spaces, insufficient exercise, and shortened circadian rhythm changes. However, the poor palatability of processed and packaged space food causes the astronauts to eat less. Astronauts, on average, eat 25–30% less than they did before flying [30]. Studies have shown that microgravity conditions do not change the amount of metabolic energy (i.e., nutrients that enter the blood through intestinal cells for use by cells) required to stay healthy [99]. Although body water loss occurs during space flight, this can only explain part of the decline in body mass [101]. The decrease of energy intake is the main cause of negative energy balance. A comprehensive review of anorexia in spaceflight suggests that microgravity during spaceflight leads to increases in the two hormones (Leptin [85] and GLP-1 [56,86]) that cause satiety. These changes in appetite-related hormones may cause the decrease in appetite observed during spaceflight. Other biological factors also affect appetite, such as astronauts’ preference for carbohydrates rather than fats [81,82].

During spaceflight, reduced food intake [81,82,83] and impaired anabolic responses [215] may reduce the production of reactive oxygen species (ROS) in mitochondria [57], and this further leads to aging, disease, and cell death [57,58]. Chronic under-intake causes permanent damage to the body [58].

While exercising in space is a popular idea for reducing muscle and bone loss and cardiovascular cleansing, it increases total energy expenditure, necessitating a greater energy intake to maintain energy balance. Exercise may further affect eating behavior, leading to acute anorexia, which can exacerbate anorexia [30]. Fresh, tasty food may stimulate the astronauts’ appetite to make them eat more. Nutrient-dense food can also help to increase energy intake more efficiently through eating.

#### 3.3.2. Effect of Microgravity

In the presence of microgravity, the energy cost of daily activities is greatly reduced due to the waste of muscles and the reduction of exercise cost in microgravity [45]. Responses to microgravity include fluid redistribution, reduced plasma volume, rapid loss of muscle mass and strength, cardiovascular deconditioning, impaired aerobic exercise capacity, bone-loss, immune and metabolic alterations, as well as effects on the central nervous system [87,88]. Muscle and bone atrophy and loss of cardiovascular function, which characterize the aging process, occur 10 times faster in space than on Earth due to microgravity-induced physiological changes [89].

#### 3.3.3. Long-Term Radiation

Space radiation can cause harmful effects such as DNA damage [90,91] and cell aging [92]. The higher cardiovascular risk among Apollo astronauts is presumed to be associated with exposure to severe deep space radiation [93]. Oxidative stress induced by space radiation and microgravity is an important factor leading to aging and disease [94]. The main measure for astronauts to resist deep space radiation is to rely on space protection facilities. Functional foods rich in anthocyanin and Omega-3 fatty acids have been used to slow down the damage caused by radiation.

#### 3.3.4. Metabolic Stress

Space missions cause metabolic stress among astronauts. Metabolic stress affects major body systems, increases the metabolic rate, and suppresses the immune system. Metabolic stress is also a strong predictor of type 2 diabetes and cardiovascular disease [93]. In addition, associated oxidative stress and inflammation have recently been implicated in the process of muscle atrophy [94] and bone loss [104]. Spaceflight has a short-term impact on the body’s iron metabolism and can lead to iron deficiency anemia as a long-term effect [61].

#### 3.3.5. Changes in Physical Condition

Astronauts often stay in bed because of their limited mobility and reduced exercise. As a result, astronauts are similar to the general population of sedentary, inactive adults and people who are confined to bed [62]. At rest, the astronauts’ core temperature will be 1 °C higher than on Earth [63]. This can be attributed to an impaired convective heat transfer and evaporation process to cool down the body, low-grade pro-inflammatory responses to weightlessness, psychological stress-induced hyperthermia, and strenuous exercise protocols leading to the so-called “space fever” [63,64,65].

#### 3.3.6. Intestinal Microecology Disorder

Microgravity leads to decreased beneficial and increased harmful bacteria in the intestinal flora. The gastrointestinal function changes accordingly, affecting the digestion and absorption function of the human intestinal tract over time [66]. In the spacecraft’s sealed living environment, relatively common infections (such as epidemic cerebrospinal meningitis, penicillin resistance staphylococcus genus) may also endanger the life of astronauts and threaten the mission [67,188].

#### 3.3.7. Vision Damage

Astronauts’ eyesight changes after landing [103]. More than half of American astronauts have a refractive change in their eyes after a long spaceflight. Findings have also included structural changes in the eye and signs of increased intracranial pressure [68].

#### 3.3.8. Fluid and Electrolyte Imbalance

Changes in liquid and electrolyte balance due to short-term exposure to microgravity have been observed in the past [216]. Such long-term and sustained changes may adversely affect the health of the crew and thereby jeopardize the success of the mission.

## 4. Discussion

Improper nutrition refers to insufficient or excessive nutrition. Lack of nutrition will lead to the risk of not being able to live and work healthily. In addition to the traditional deficiencies of protein, vitamins E, K, D, polyphenols, and polyunsaturated fatty acids, deficiencies of minerals (calcium and potassium) and low elements (iron) have been observed in long-term space missions. [53]. Excessive intake of certain nutrients may lead to symptoms such as scurvy and beriberi [69]. Improper nutrition is not associated with caloric intake. During the European mission at the end of the last century [217,218], astronauts had an inadequate intake of energy, liquid, and calcium in addition to excessive sodium compared to the dietary reference intake on Earth. Inappropriate levels of these nutrients have considerable effects on hormonal regulation, cardiovascular functioning, and calcium and bone metabolism. Efficient space nutrition should function as a countermeasure for the effects of the space environment on astronauts’ physiology and metabolism [30,84]. Therefore, space nutrition should not only provide astronauts with sufficient energy intake but also allocate nutrients to counteract the adverse effects of the space environment [83]. This section outlines possible future research directions for space nutrition as countermeasures for the effects of adverse space living conditions.

### 4.1. Nutritional Measures to Cope with Reduced Intake

This section outlines possible future research directions for space nutrition as countermeasures for the effects of adverse space living conditions.

#### 4.1.1. Increase Palatability through Fresh Food with Distinctive Flavors

The reduced intake of space food by the astronauts is potentially because their sensory responses to the taste, smell, and texture of food in space [50] reduce the perceived attractiveness of food [45,99,100]. Even though the quality of food has improved since the beginning of the space program, food provided to astronauts is still not as palatable as what is available on Earth. At the same time, the artificial fortification of food can lead to the destruction of nutrients over a long period of time and can also produce an unpleasant taste [31]. Processed food has major limitations. The 3-to-5-year shelf life limits the variety of processed food, and many processed food items lose their original flavor. The traditional focus on the form of food and meeting the crew’s caloric needs is no longer sufficient for long-term missions [31]. There is an urgent need for fresh food with distinctive, original flavors. Future research should investigate ways to maximize fresh food production despite the space, weight, and resource constraints during space flight.

#### 4.1.2. Boost Energy Intake through Dietary Culture and Food Production Activities

The stress astronauts experience during spaceflight alters their appetite and energy intake [217,218]. Improvements in the variety and quality of food and the emphasis on dietary culture may help boost their energy intake [45]. Dietary culture involves the use of familiar food sources and utensils, cooking methods, and eating methods and places. Meals play an important role in people’s emotional and social wellbeing. Having astronauts participate in the production, harvesting, and cooking of ingredients not only increases the amount of exercise for them but also relieves stress and induces happiness. The happiness induced can be mainly attributed to the sense of fulfillment one obtains from participating in the agricultural process. Further, cooking and sharing food together helps astronauts identify with the space environment and feel a sense of belonging. The resultant place identity and attachment increase the astronauts’ food intake, helping to maintain their physical health. In space exploration missions, the creation of human habitats around food production areas as loci of emotional attachment and social interactions is paramount to the success of long-term missions.

#### 4.1.3. Enhance Caloric Intake through Nutrient-Dense and Fresh Foods

The type of food and the amount of nutrients in the diet can also affect caloric intake [45]. Astronauts often prefer carbohydrates rich in energy and micronutrients [81,118]. However, the supply of such low-density food is limited, because the space environment presents unique challenges for transporting large quantities of food. If carbohydrates are used in large quantities to provide space nutrition, large quantities of drinks and vegetables will also be required [31].

Nuts are by far the most effective food, with relatively high energy and important nutrients in a compact food matrix. However, nuts are rich in fat and protein, both of which can affect shelf life. Future space nutrition research should address significant gaps in supplying more lightweight nutrient-dense foods [31] and developing fresh food production methods to better meet the astronauts’ demand for a large amount of carbohydrates, vegetables, and drinks.

#### 4.1.4. Counterbalance the Effects of Space Environment on Leptin Secretion with Nutrition

The high concentration of carbon dioxide in spacecrafts and the ISS changes energy intake and reduces food consumption [70,71]. Astronauts’ eating habits are affected by the 90-min diurnal cycle in the Earth’s orbit and the insufficient light level in spacecraft [51,72]. The ambient light cycle variations disrupt the circadian rhythms and affect the hormonal balance in the body [73,74,75]. For example, changes in light and dark cycles have been shown to change the rhythmicity of leptin secretion and affect food intake and body weight in rodents [76]. Thus, continuous light exposure may at least partially explain the higher leptin levels in astronauts [56,72]. To mitigate the effects of excess leptin, space nutrition research should investigate ways to reduce inflammatory foods, such as sugary drinks and trans fats while increasing anti-inflammatory foods, such as olive oil, avocado, coconut, fish, and grass-fed pasture-raised animals. Long-term space habitats should be designed to regulate circadian rhythms and facilitate the production of anti-inflammatory foods. To overcome the payload limitations at launch while providing non-plant-based proteins, future research should investigate ways to use in-space manufacturing to construct large-scale space habitats with artificial gravity to accommodate growing at least fish, if not animals, from embryos.

#### 4.1.5. Improve Immunity with Nutritional Measures

Astronauts had different immune responses to a decrease in their energy and nutritional intake [69]. For example, a decrease in mitogen proliferation reaction is related to deficiency in Vitamin B6 (VB6), Vitamin B-12 (VB12), biotin, Vitamin E (VE), copper or selenium. Delayed hypersensitivity is associated with VB6, VB12, VC, or iron deficiency. Protein and individual amino acid deficiencies have different effects on various immune functions [84,119]. Providing effective nutrition for astronauts can help to combat immune dysfunction during spaceflight.

### 4.2. Nutritional Countermeasures for the Effects of Microgravity

There are a wide range of health risks associated with the side effects of adapting to weightlessness during space flight. These side-effects include changes in musculoskeletal, cardiovascular, neural vestibular, immune, endocrine, and other physiological systems. Muscle catabolism and bone loss are two physiological changes that occur in weightlessness. The use of exercise and medication has limited success in mitigating these side effects without an appropriate diet to mitigate the deleterious effect of microgravity [83,120]. A recent study shows for the first time that adequate energy, protein, and vitamin D supplies are needed to maintain bone density after a 6-month spaceflight [103].

#### 4.2.1. Mitigate Bone Loss with Nutritional Measures

Bone loss is one of the important factors affecting astronauts’ health. Although its mechanism is not fully understood, it is clearly multifaceted. Dietary intake resists changes in bone metabolism during space flight. An optimized diet [105,106] helps to mitigate bone loss with little risk of side effects by providing adequate amounts of calcium, vitamin D, and vitamin K in the space diet [107]. Many parallels have been found between nutritional orthopedics and space nutrition because of the similar challenges faced by an aging orthopedic patient on Earth and an astronaut experiencing the degenerative effects of long-duration space missions [108]. A daily balance of fiber, liquids, and bioactive substances, such as coffee, is necessary to prevent hip fracture when transitioning from a low-gravity field to hyper-gravity while landing on a planet.

#### 4.2.2. Reduce Sodium Intake

Most space food has a high sodium content, mainly sodium chloride. This leads to a high dietary sodium intake for astronauts, averaging more than 5000 mg/day and more than 12,000 mg/day for individuals [53,55,82,132,133]. High sodium chloride intake increases urinary calcium excretion [134,135,219,220] and the risk of kidney stones [136], as well as low-grade metabolic acidosis [137,138], which further leads to increased bone resorption [138]. A reduced intake of sulfur-containing amino acids and sodium chloride has been shown to reduce bone loss during bed rest [109]. NASA has made an enormous effort to reformulate 90 foods to reduce sodium intake to 3000 mg/day [109]. As sodium chloride is a food additive used in space food processing and preservation, it is challenging to reduce sodium content without negatively affecting the taste of food. The ultimate solution to excess sodium intake is to replace packaged food with fresh food.

#### 4.2.3. Increase Intake of Vegetable Protein, Potassium, and Bisphosphonates

The ratio of protein to potassium in the diet may also affect bone metabolism [109,110]. The type of protein in the diet is also important for bone health [103]: animals have lower potassium (and potassium salts) than plants. Animal proteins are usually high in sulfur-containing amino acids, and animals have lower potassium than plants. The oxidation of sulfur-containing amino acids may lead to low-level metabolic acidosis and corresponding bone resorption which can be compensated by reducing the animal protein–potassium ratio, especially in the final stages of bed rest studies [103]. The arrhythmias in the Apollo 15 astronauts were caused by a lack of potassium due to a lack of nutrients in the space food system [110]. The potassium deficiency in this short-term task was alleviated by potassium supplementation in the subsequent task [31]. Potassium citrate, as a non-sports antagonism scheme for renal calculi, has been transferred from the flying observation stage to the clinical research stage. In addition to vegetables that are rich in plant protein and potassium, supplementation with bisphosphonates may be a countermeasure to bone loss [122,123]: potassium-rich foods include legumes, peanut, mushroom, laver, and kelp. Plant protein mainly comes from various forms of beans.

#### 4.2.4. Increase Vitamin D Intake

Adequate energy and vitamin D intake, as well as regular resistance exercise, can significantly improve bone condition [103]. A space nutrition system low in vitamin D, coupled by the lack of ultraviolet radiation, leads to vitamin D deficiency in the astronauts [47]. Inadequate intake of vitamin D supplements can cause body reserves to lose vitamin D during flight [53,111]. While vitamin D may not be the cause of bone loss during space flight, it certainly exacerbates the problem. Foods that contain more vitamin D include fish, milk, liver, eggs, mushrooms, and beef. It is imperative for long-duration space flights to develop effective means to cultivate mushrooms because it is not feasible to produce fresh fish and animal protein in the near future due to weight and space limitations associated with the current space flight capabilities.

#### 4.2.5. Increase Vitamin K Intake

Newborns, pregnant women, and astronauts tend to experience deficiencies in vitamin K, which is known as the clotting vitamin. There are two kinds of natural vitamin K: (1) vitamin K1 promotes blood clotting for female astronauts to help reduce period hemorrhage in great quantities and prevent internal hemorrhage and hemorrhoid [124]; (2) vitamin K2 is involved in bone metabolism. Studies have found that loss of vitamin K is associated with low bone mass or increased bone loss [125]. Supplementation with vitamin K has been shown to normalize low carboxylate levels of osteocalcin, counterbalancing the loss of bone formation [126,127] and helping to protect astronauts from bone loss. At the same time, vitamin K can promote the absorption of calcium elements to effectively prevent bone marrow calcium loss.

Vitamin K-rich foods include yogurt, alfalfa, egg yolks, fish eggs, algae, carrots, and green leafy vegetables. The roots of Gynura bicolor (*Begonia fimbristipula* in Latin) have been used as a functional food in aviation because they contain vitamin K, which is absent in most vegetables. It has been developed as a functional food in aviation [128]: after the human body absorbs the vitamin K1 in Gynura bicolor, the vitamin K1 will be partially converted into the vitamin K2. This plant has a significant application value in addressing calcium and bone loss for the astronauts because it can effectively improve their physiological function through promoting the rapid normal coagulation of blood and increasing the absorption of calcium to minimize bone loss. In addition, Gynura bicolor helps to protect astronauts from accidental injuries while enhancing the wellbeing of female astronauts during menstruation.

#### 4.2.6. Increase Calcium Intake

The space environment causes bone loss, which in turn, causes calcium loss. The time spent in the adverse space living environment is associated with the amount of bone and calcium loss. In-flight astronauts’ calcium intake was often close to or at the planned intake, similar to the recommended intake on Earth [47,53,127] despite reduced absorption, increased urinary calcium, and a further risk of kidney stones [109]. Calcium-rich foods include milk, legume, fish, shrimp, kelp, black agaric, laver, and sea cucumber. Increasing the intake of fruits and vegetables may prevent changes in bone mineral content when adequate calcium is consumed during spaceflight [107].

#### 4.2.7. Increase Unsaturated Fatty Acids and Decrease Saturated Fatty Acids

Unsaturated fatty acids are superior to saturated fatty acids in preventing metabolic changes, including insulin resistance and inflammation [140,141]. Studies have shown that omega-3 fatty acids [120] and vitamin E [142] help facilitate space adaptation [142]. Omega-3 is a family of polyunsaturated fatty acids, including three types of fatty acids: namely, α-linolenic acid, DHA, and EPA, which are found in linseed oil, vegetable oil, fish oil, seaweed, and leafy greens. Omega-3 contains essential nutrients that cannot be synthesized by the body and must be obtained from food. omega-3 intake has been shown to reduce bone resorption during bed rest [120]: to obtain plentiful omega-3, fish is the best animal resource, while flax and peony seed oil are the best plant sources. Fruits and vegetables also play an important role in providing omega-3. Providing a diet rich in fruits and vegetables, such as a Mediterranean diet, over the long term will have a positive impact on bones in long-term tasks, while adding omega-3 and fish can help counteract bone loss. Fish and omega-3 have also been found to support bone health protection [143,144], mitigate muscle atrophy, and enhance immune function [132]. Thus, an increase in the intake of fish may help strengthen the skeletal muscle to support a strong immune system in addition to reducing the risk of cancer [145,146,147,148,149,150]. Micronutrients in nutrient-rich foods, such as fruits or vegetables, can also help to combat most physiological disorders caused by microgravity [189].

#### 4.2.8. Increase Protein Intake to Counteract Muscle Atrophy

Exposure to microgravity reduces muscle mass, volume, and performance [47]. Weightlessness can lead to changes in skeletal muscle function and structure, such as decreased muscle mass, fatigue, motor nerve stiffness, and autonomic motor ability. Therefore, it is particularly important for astronauts to maintain at least their normal energy supply and protein intake for short-term space missions.

Studies of astronauts on long flights that last more than 100 days suggest that a decrease in protein synthesis may be related to a lack of energy intake [129]: the data showed that the protein macronutrient compositions at an average of 15%, as recommended by the World Health Organization (WHO), were underestimated by 41%. Therefore, astronauts should intake more protein for long-term space missions. In addition to protecting astronauts’ health, protein is also crucial to their ability to move when they return to earth.

Supplementation with essential branched chain amino acids enhances protein synthesis to help fight muscle atrophy [130,131]: branched chain amino acids stimulate the production of insulin, which also promotes the absorption of amino acids by muscles. Branched chain amino acids also help prevent protein breakdown and muscle loss. Foods containing branched chain amino acids are fish, shrimp, milk, soy, corn, glutinous rice, cauliflower, and so on.

Supplemental animal protein has been shown to reduce the pH of the blood through the oxidation of sulfur amino acids, thus exacerbating bone loss [117]. Astronauts should limit the amount of animal protein they eat to minimize bone loss while increasing other sources of protein to stay above their normal protein intake level for long-term space missions.

#### 4.2.9. Combat Intestinal Microecology Disorder with Foods Rich in Calcium and Probiotics

Astronauts often suffer from a variety of infectious diseases caused by fungal hypersensitivity or the degradation of protective intestinal flora [188,216]. To combat the microgravity-induced degradation of intestinal flora, Italian space agency researchers developed calcium- and probiotics-rich space food from freeze-dried plain yogurt and uniquely flavored yogurt with blueberries [131].

### 4.3. Nutritional Countermeasures for the Effects of Radiation

One of the greatest hazards of space flight is space radiation. Space radiation can cause acute physical damage to human tissues, such as skin bowel marrow and other tissue, leading to cataracts. Radiation can also kill human cells and change the DNA in the body, reducing immunity and increasing the risk of cancer and genetic damage [90,91,92,93,94].

Food does not prevent direct damage from space radiation, but it can help astronauts tolerate low levels of radiation exposure. To prevent or reduce radiation damage from the indirect effect of radiation that produces free radicals, countermeasures can be provided through a diet rich in the following ingredients: natural antioxidants (such as procyanidins), Omega-3 fatty acids, vitamin E, vitamin C, beta-carotene, selenium, and even plants containing dietary fiber. Foods with antioxidant properties include tomatoes, garlic, nuts, oats, blueberries, broccoli, salmon, and green tea. Tonic food rich in a certain vitamin helps enhance the health performance of the human body. One example of such food is buckwheat, which contains high amounts of Vitamin P and selenium. Selenium is an active component of the antioxidant glutathione peroxidase, which can effectively remove free radicals, fight oxidation, and activate the immune system to prevent cancer. Vitamin P can effectively inhibit bacteria and the carcinogenicity of aflatoxin B1. At the same time, selenium can catalyze and eliminate free radicals harmful to the eyes and treat a variety of eye diseases, such as cataract retinopathy. Therefore, foods rich in selenium are promising aviation functional foods.

### 4.4. Brief Summary of Nutritional Countermeasures to the Adverse Effects of Space Environment

Different health problems are caused by insufficient calories and micronutrient intake. Targeted nutrition countermeasures should be provided to meet the necessary nutritional needs of crew members to prevent the occurrence of diseases to better sustain space missions. While VB6 is most abundant in yeast, it is also found in wheat bran, malt, liver and kidney, rice, potatoes, sweet potatoes, vegetables, carrots, bananas, and peanuts [142]. VB12 is found in shellfish livers and in all animal sources, such as fish, shrimp, eggs, and milk, as well as a few non-animal sources, such as fermented soy products, mushrooms, and seaweed [143]. VE is widely found in nuts, lean meat, milk, egg, vegetable oil, wheat germ, green leafy vegetable, sweet potato, yam, and kiwi fruit [144]. VC is widely found in fresh vegetables and fruits [145]. The most ideal sources of biotin are yeast, liver, kidney, brown rice, peanuts, beans, fish, and egg yolks [146]. Iron is concentrated in animal liver, clam, kelp, shrimp, chicken with egg yolk, beans, green vegetables, and fruits [147]. The main way for the human body to supplement copper is to eat animal liver, shellfish, fish, meat (especially poultry), fruits, tomatoes, green peas, potatoes, shellfish, laver, cocoa, and chocolate [149,150,189]. Selenium is most abundant in seafood shellfish, followed by animal viscera and kidney wheat germ. An insufficient energy intake can affect protein turnover, thus increasing protein and individual amino acids. In summary, VB12, biotin, copper, and selenium cannot be sourced from vegetables, which are the only types of fresh space food currently. It is essential for long-term space habitats to support the production of animal proteins, fermented soy products, seaweed, mushrooms, and brown rice with minimal weight, resource, and space requirements. Table 3 summarizes the nutritional countermeasures for the adverse effects of the space environment.

### 4.5. Nutrient Loss during Food Processing and Storage

For long-duration missions, the risk of malnutrition for astronauts comes from the inadequate quantity and quality of the food they intake. Currently, the food items consumed in space are largely processed and stored foods with limited shelf life. For most space foods, NASA requires a shelf life of 18 to 24 months, which is not enough for future deep space missions. Long-term missions require the food shelf life to be extended to 3 to 5 years [221]. It is a challenge to maintain the quality of most space foods within a span of 5 years. If the food loses quality after the first 2 years, the crew will not get enough nutrition for space missions longer than 2 years even if the food does provide sufficient nutrition for the first 2 years [31].

Astronauts have suffered from intaking space foods with varying degrees of texture and nutrition loss due to the following [31]: thermal sterilization during processing, vibration and shock during storage and transportation, and radiation accumulation during long-term storage in space. Existing data show that storage temperature and time have a significant impact on the vitamin content of space food.

The results of an experiment conducted with space food storage over 2 years provided the following insights: (1) when the storage temperature of several fruits and vegetables was 27 °C, the loss rate of VC, VB1, and VB2 was up to 58%; (2) when the storage temperature was 10 °C, the maximum loss rate of vitamin was 38% [84,151,153]. Because of its long shelf life, thermally stabilized food is the main type of long-term manned spaceflight food. Thermally stabilized foods undergo destructive heat treatment processes that result in nutrient loss, flavor deterioration, and other changes in food quality. The flavor of food changes with time, mainly manifesting as rancidity, a low aroma, and an overall decrease of flavor [38,84,221].

Over 2 years of food storage is significant for nutrient loss in most foods [206]. According to an experiment conducted to study the degradation of vitamins during storage, there was significant reduction in folate, thiamine, VA, VC, and VK in various space foods for a period over 880 days. After 596 days (about 1.5 years), significant differences in vitamin concentrations were found in tacos, salmon, and roasted broccoli. At the end of the study, the multivitamin also underwent chemical degradation, and riboflavin, VA, and VC were reduced by 10% to 35%. Space radiation has no effect on the nutritional level of any food [31]. Cooper (2010) stored foods at 22 °C over 5 years and analyzed 24 vitamins and minerals in foods at different times over 1 year, 3 years, and 5 years. The preliminary conclusion is that the thermal stability of foods leads to the degradation of VA and VC, thiamine, and folate, and subsequent oxidation further promotes the degradation of vitamins in storage. In most product packaging, VA continued to decrease during the first year of storage. Similarly, most folate and thiamine levels dropped, and VC levels in all products dropped from 37% to 100%. Nutrient loss at 3 years to 5 years is thus expected to be significant and may lead to undernutrition in the food system. As a result, nutrient loss over 3 to 5 years is expected to be significant and may lead to a decrease in the nutritional contents of the food system [153].

Conventional methods of storage have failed to maintain the initial quality of food for long-term missions, necessitating research into developing new processing techniques and storage conditions. New methods of food supply may also need to be considered.

### 4.6. Hazards of Food Packaging and Additives

Packaging plays a vital role in the food supply chain by protecting food from environmental influences such as oxygen, water, light, dust, pests, and volatile matter, as well as chemical and microbial contamination [166,171,172,222].

#### 4.6.1. Toxicity of Packaging Materials

The packaging forms and basic materials of food used for consumption in space mainly include the following four types [166,171,172,222]: (1) frozen food packaging using basic materials, such as plastic (OPP, NY, LLDPE, PET, PE), paper, and composite aluminum foil; (2) semi-solid food packaging and hose packaging, including aluminum foil, plastic, and resin; (3) canned packaging using aluminum foil, plastic film, and kraft board; and (4) packaging in bags, aluminum foil, and plastic. In summary, the main food packaging materials are aluminum foil, plastic, resin, and glass, among which aluminum foil and plastic are the most widely used materials.

In recent years, processed and packaged foods have been identified as some of the main sources of human exposure to plasticizers and bisphenols, which have migrated from plastic packaging [154]. Phthalates are used as plasticizers in many consumer products, mainly in food packaging, flooring, household, and other consumer products [223]. Bisphenol A (BPA) is used primarily in the production of polycarbonate plastics and epoxy resins, as a protective coating in food cans, and in water and feeding bottles. Neither phthalates nor BPA chemically bind to products. In addition, they penetrate, migrate, or evaporate into indoor air, dust, the atmosphere, and food. Phthalates have attracted public attention over the past few decades because of their potential health risks [154]. Dibutyl phthalate (DBP), Benzyl butyl phthalate (BBP) and Diethyl p-nitrophenyl phosphate (DENP) are listed as compounds with suspected endocrine disrupting functions. The serious effect of bisphenol compounds on liver and kidney function was confirmed in mice. BPA can affect liver and kidney function even at very low doses [155,156,157,158]. Surprisingly, plasticizers, especially BBP and bisphenols, were detected in high concentrations in foods. Their contamination is widely involved in packaged foods including grains and cereal products, milk and dairy products, fats and oils, fish, and candy. Foods that are high in fat are most affected by BBP [159,160,161].

Aluminum foil packaging is a composite material made of various materials and aluminum foil, which is a thin metal film made directly from aluminum. With the extension of temperature and time, acid food and alcohol can cause significant aluminum dissolution in packaging foods. Aluminum dissolves and causes heavy metal pollution to food, harming the human body [162]. The toxicity of aluminum is manifested in brain, liver, bone, hematopoietic (multipotent) stem cells, and cells [163]. The health of space residents can only be achieved by eating raw, natural fresh foods without the long-term, slow toxicity from packaging materials.

#### 4.6.2. Health Threats from Food Additives

Long-term space missions require a shelf life of up to 3 to 5 years for food, especially for ingredients that are high in oil and easily oxidized. Packaged food contains food additives, which are (1) synthetic or natural substances added to food for the purpose of improving the quality of food such as color, aroma, and taste, as well as (2) anti-corrosion and processing technology. It remains challenging to preserve food for long-duration missions without a wide range and a significant number of preservative-active substances or carriers. The U.S. space agency recently reduced sodium in food by 90% [103]. Many food additives are bound to threaten the health of astronauts.

#### 4.6.3. Challenges Associated with New Packaging Technology

There has been a growing demand for unprocessed or minimally processed natural high-quality foods that are preservative-free but have an acceptable shelf life [164,165]. The protective functions of packaging have been refined and improved through the development of new packaging technologies as follows: (1) modified air packaging (MAP) [167,168,169,170], (2) active packaging (AP) [173,174,175], (3) intelligent packaging (SP/IP) [174,176,177,178,179], and (4) the use of nanomaterials for packaging [182,183,184]. However, these packaging systems can only extend the shelf life of packaged foods by a few days to a year [185]. This is far from the 3–5-year shelf life necessitated by long-term missions. In addition, these packaging systems have several disadvantages, such as risk of accidental breakage, the need for additional packaging procedures, and moisture sensitivity, making them unsuitable for use with beverages or wet food [166,186,187]. Above all, space missions cannot rely entirely on packaged food.

### 4.7. Development of Space Food Systems

There are currently two lines of inquiry for developing space food production systems for long-term space missions. One is centered on small-scale aseptic food production systems in weightless settings, such as the ISS and most transit space habitats that cannot produce their own artificial gravity. The other investigates the establishment of large-scale closed-loop ecosystems within planetary surface space habitats on the surface of the Moon and Mars.

#### 4.7.1. Aseptic Food Production Systems for Transit Space Habitats

To effectively use space nutrition as a countermeasure for microgravity-related health risks, the production of fresh foods is necessary to enhance nutrient diversity and dietary acceptability for the astronauts [42,188,191,192,193,224]. Since major crops can be dried and preserved for a long time, the production of fresh fruits and vegetables with hydroponic, aeroponic, and substrate culture techniques helps to enhance the food quality and the variety of the menu to ensure that astronauts can acquire sufficient quantities and types of natural vitamins [194,197].

Microgravity requires different approaches to store and circulate nutrient solutions to the root zones of plants in small chambers, motivating the development of many closed-looped small-scale food production systems: the NASA Ames research center has developed a “salad machine” that provides astronauts with fresh salad vegetables such as lettuce, cucumbers, and carrots [201]. In April 2014, a vegetable production chamber called “VEGGIE” arrived at the ISS. It became the first system to grow plants on the space station. The VEGGIE project is the first step towards building a “self-sufficient” space station. VEGGIE also provides “gardening therapy” for astronauts to relieve stress and depression [202]. NASA introduced LED growth lights for growing tomatoes and fresh salad greens. The team adjusted lighting conditions to optimize plant growth under a variety of conditions and then replicated these scenes in Advanced Plant Habitats on the ISS to meet the complex requirements of producing food in space [203]. In 2016, the U.S. applied the cultivation method of the “Plant Pillow” (substrate culture technology) to grow crops in space [205]. In 2017, the ISS started to produce fresh food with a self-watering hydroponic system called the Passive Orbital Nutrient Delivery System (PONDS); the system uses capillary mat materials to connect the substrate around plant roots with a reservoir containing oxygen-infused nutrient solutions [207]. Tupperware and Techshot have subsequently developed a semi-hydroponic system that requires less maintenance.

There are many challenges associated with food production and preparation in microgravity. For example, if fresh fruits and vegetables are eaten without proper cleaning protocols, they can be vulnerable to microbial contamination. Eating non-sterile food presents astronauts with some level of risks because the sterilization procedures that work on Earth may not be as effective in space, where microbes can mutate more quickly and exhibit more robust behaviors in microgravity [208]. A proven method for monitoring the microbiome of fresh space foods is needed to reduce potential health risks associated with potentially harmful food pathogens. Microgravity and the lack of atmospheric pressure also affect the transfer of heat and mass, making it challenging to use common cooking methods to prepare food in space [31].

Many crops have been successfully grown in microgravity [209,210]: onions, cucumbers, and radishes were cultivated on Salyut 7. The vegetables currently grown by astronauts include tomatoes, strawberries, and lettuce. These crops were included because they have a short growth cycle in space and ripen quickly enough to be edible after 28 days. The astronauts ultimately expanded their selection of edible plants to include a wide range of fruits and vegetables, including wheat, peanuts, soybeans, mizuna, pea, and other food crops. Space nutrition became increasingly more closed-looped as the astronauts’ metabolic wastes were converted into sources of nutrients for food production or even processed food through biotechnology [211].

#### 4.7.2. Long-Term Food Production Systems as Closed-Loop Life-Support Systems

With perchlorate at levels commonly toxic to plants, the Martian regolith cannot become a viable growing substrate without addressing restricted metal availability and nutrient enrichment [212]. However, even with nutrient supplementation, none of the three currently available Martian Regolith Simulants (MRSs) are capable of supporting plant growth [213]. These studies suggest that closed-loop life-support systems will be necessary for supporting food production on Mars. Similarly, the lunar regolith cannot support plant growth effectively without addressing issues associated with potentially toxic elements, pH, nutrient availability, air and fluid movement parameters, and its cation exchange capacity [214]. Compared to sand and sungro, pioneer plants, such as *Sporobolus airoides*, have been found to germinate equally well in a new agricultural grade lunar regolith simulant (General Lunar Nearside Highland Non-Agglutinate Regolith Simulant) developed by Off planet Research, LLC, and Saint Martin’s University [198]. Supplementing closed-loop life support systems with mechanisms for resource separation will help maximize in-situ resource utilization without introducing toxic elements into closed-loop life support systems.

The German aerospace center has developed an advanced closed-loop life support system that uses algae-powered photobioreactors to provide continuous and breathable air in space by converting astronauts’ breath and sunlight into oxygen and food. The system could provide about 30% of the food that an astronaut needs. The concept of the human habitat is based on a symbiotic relationship between humans and plants. Edible plants consume carbon dioxide and release the oxygen that humans need. In return, human waste and non-biodegradable plant matter energy (after decomposition in a microbial tank called a bioreactor) provide nutrients for plant growth. These plants can also provide medicine. However, the effects of gravity, light, atmosphere, soil, and radiation on the growth capacity of plants in this regenerative life support system remain largely unknown [197].

On the surface of the Moon and Mars, higher plants from regenerative life support systems (controlled ecological life support systems) will be the main food source for astronauts. They include wheat, potatoes, sweet potatoes, peanuts, soybeans, rice, and dried beans. Salad ingredients include tomatoes, onions, spinach, beets, cabbage, carrots, lettuce, and radishes. Food processing equipment suitable for the space environment should take into account safety, power, volume, water consumption, air pollution, waste production, cleanliness, and noise. One such piece of equipment is a multifunctional fruit and vegetable processing system developed by NASA for cutting, heating, separation, and concentration [199,200]: for example, the system can process tomatoes into tomato juice, tomato sauce, tomato slice, and tomato soup.

## 5. Conclusions

The first major finding suggests that resupplies for long-term space mission are cost-prohibitive although it is technologically feasible for short-term missions. It would be even more challenging and costly to resupply food from the Earth to the Moon or Mars as a destination for roundtrips or long-term stays. The finding implies the need for long-term space nutrition system to rely on self-sufficient bioregenerative systems to produce fresh foods. Such systems can be implemented in situ on the Moon or Mars or in upscaled space habitats with artificial gravity.

The second major finding reveals the inadequacy of existing space nutrition systems for meeting long-term physiological and psychological needs due to the dominance of processed and functional foods. Fresh food materials provide natural vitamins, minerals, dietary fiber, and secondary compounds that are lacking in packaged foods. Packaged food cannot meet the health needs of astronauts during long-term space missions due to the loss of nutrients during food processing, preparation, and storage, as well as the health risks associated with using packaging materials and food additives. At the same time, ready-to-eat space food cannot meet the astronauts’ psychological needs for (1) a sense of familiarity from undertaking their normal eating habits and maintaining their food culture and (2) a sense of community from engaging in food production, preparation, and consumption as social activities [190]. Therefore, space missions cannot rely largely on packaged food to help astronauts cope with the adverse space environment. While functional food may help to counter some of the aforementioned adverse living conditions in space, it is not economically feasible to carry sufficient functional food for long-term space missions lasting 3 to 5 years. Fresh food production through the use of closed-loop systems remains necessary for providing a diverse range of nutrients while working within the limited payload threshold allowed by each space flight. There is an urgent need for the sustainable production of fresh food with distinctive, original flavors to increase appetite. During long-term space missions, enabling a self-sufficient lifestyle through the use of a nature-based regenerative life support system will help astronauts better adapt to the adverse conditions of the space environment [195,196]. This landscape-based approach to space habitat design helps engender a sense of place attachment from biophilia; that is, the human instinctual attachment to life-like features found in nature. Place attachment can potentially make astronauts more resilient in space psychologically and physiologically to better ensure the success of long-term space missions.

The third major finding is that the current small-scale food production systems used by the ISS to produce fresh food are inadequate for completely replacing packaged food without drastically upscaling the next generation space stations and habitats for long-term space missions to the Moon and Mars. It is necessary to undertake a modular approach to increase the size of transit space habitats for long-term space missions through in-space manufacturing. The generation of artificial gravity will be important to safeguard food safety and crew health by enabling the use of more ecosystem-like closed-loop life support systems without increasing the safety risks associated with a higher likelihood of microbial contamination in microgravity. As artificial gravity and in-space manufacturing become increasingly more feasible, incorporating nature-based environments within a landscape-scale space habitat will be a critical path to providing sufficient fresh food to space inhabitants as not only a form of complex medicine required for long-term psychophysiological wellbeing and but also indispensable countermeasures to the adverse living conditions in space.

## Figures and Tables

**Figure 1 nutrients-14-00194-f001:**
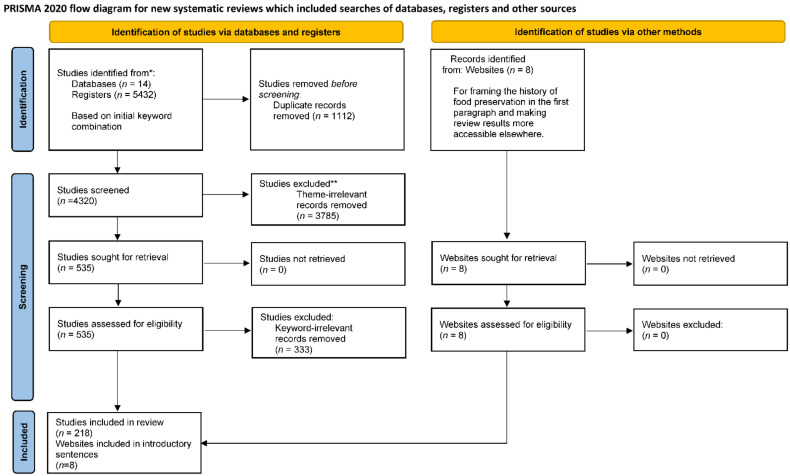
Review protocol for searches of databases, registers, and other sources.

**Table 1 nutrients-14-00194-t001:** Relevant references for the final list of keywords as eligibility criteria.

Keywords	Reference Numbers
Long-term space tasks	[9,10,11,12]
Space food systems	[30,31,32,33,34,35,36,37,38,39,40,41,42,43,44,45,46,47]
Diet menu fatigue	[31,45,48,49,50,51]
The impact of space environment on astronauts	[14,15,16,17,19,30,45,47,51,52,53,54,55,56,57,58,59,60,61,62,63,64,65,66,67,68,69,70,71,72,73,74,75,76]
Dietary and nutrition deficiencies	[30,77,78,79,80,81,82,83,84]
Microgravity	[45,85,86,87,88,89]
Space radiation	[90,91,92,93,94]
Weight loss	[45,53,78,79,80,81,83,95,96,97,98,99,100,101,102]
Bone loss	[13,14,15,47,53,99,103,104,105,106,107,108,109,110,111,112,113,114,115,116,117]
Nutritional strategies	[30,31,47,53,69,81,83,84,103,107,111,118,119,120,121,122,123,124,125,126,127,128,129,130,131]
Reduce sodium intake	[53,55,83,109,132,133,134,135,136,137,138]
Fatty acid	[120,139,140,141,142,143,144,145,146,147,148,149,150]
Nutrient loss during food processing and storage	[31,38,84,151,152,153]
Security threat of packaging materials and food additives	[75,103,154,155,156,157,158,159,160,161,162,163,164,165]
New packaging technology	[163,166,166,167,168,169,170,171,172,173,174,175,176,177,178,179,180,181,182,183,184,185,186,187]
Fresh food materials	[42,151,188,189,190,191,192,193,194]
Self-sufficient	[20,21,22,31,195,196,197]
Space habitats	[197,198,199,200]
Space food production	[201,202,203,204,205,206,207,208,209,210,211,212,213,214]

**Table 2 nutrients-14-00194-t002:** List of databases used for the systematic review.

Chinese Databases	English Databases	
Wanfang Medical Network	EBSCO	
X-MOL Information Retrieval	Web of Science (SCIE)	
CQVIP Chinese Journal	OVID and CAB Plus Full-Text	
CQVIP Chinese Biomedical Journal	PROQUEST Agriculture and Biology
CNKI Citation	Springer Link Full-Text	
BvD JSTOR The Merk Index	Oxford Journals Collection	
Doc88.com Literature Sharing Platform *	Kopernio Chrome	

* Accessed date: 5 November 2019.

**Table 3 nutrients-14-00194-t003:** Nutritional countermeasures for the adverse effects of space environment.

Issues	Nutritional Strategies		Recommended Food and Nutrition
1. Nutritional measures to cope with reduced intake	Increase the appeal of space food		Fresh food with a distinctive flavor
	Pay attention to space food culture as a source of joy		Participate in the production, harvesting, cooking, and sharing of fresh food materials with peers to build a sense of belonging
	Meet the astronauts’ carbohydrate preferences		Grow fresh vegetables and food in space to meet astronauts’ demand for large amounts of food
	Add foods with high energy density		Nuts
2. Nutritional measures to cope with decreased immune function after weight loss	Supplement VB6		Yeast, wheat bran, malt, liver and kidneys, rice, potatoes, sweet potatoes, vegetables, carrots, bananas, and peanuts
	Supplement VB12		Shellfish, livers, and all foods derived from animals. Fish, shrimp, eggs, milk, and fermented soy products
	Supplement VE		Nuts, lean meat, milk, eggs, vegetable oil. Wheat germ, green leaves, sweet potato, yam, and kiwi
	Supplement VC		Fresh vegetables and fruits
	Supplement Biotin		Yeast, liver, and kidney. Brown rice, peanut coat, beans, fish, and egg yolk
	Supplement Iron element		Liver, clams, seaweed, fish, shrimp, egg yolk, chicken, beans, green leafy vegetables, and fruits
	Supplement Cuprum		Liver, shellfish, fish, meat (especially poultry), fruits, tomatoes, green peas, potatoes, shellfish, laver, cocoa, and chocolate
	Supplement Selenium		Seafood shellfish, animal viscera, kidneys, and wheat germ
	Supplement Protein		Protein and individual amino acids
3. Nutritional measures to cope with the effects of microgravity	Mitigate bone loss	Reduce sodium	Reduce sodium chloride intake to replace stored with fresh food.
		Add vegetable protein	Increase plant protein: rice noodles, and beans Increase potassium citrate, and supplement high potassium ingredient, such as beans, peanuts, mushrooms, seaweed, and kelp
		Supplement VD	Fish, milk, liver, eggs, mushrooms, and beef
		Supplement VK	Yogurt, alfalfa, egg yolks, fish eggs, algae, carrots, and green leafy vegetables
		Supplement calcium	Milk, beans, fish, shrimp, seaweed, black fungus, seaweed, and sea cucumber
		Prioritize unsaturated Omega-3 fatty acids	Fish, flax, peony seed oil, fruits, and vegetables
	Fight muscle atrophy	Increase protein intake	Food containing branched chain amino acids: fish, shrimp, milk, soy, corn, glutinous rice, and cauliflower.
	Intestinal microecology disorder	Supplement probiotics	Yogurt rich in calcium and probiotics
4.Against radiation	Provides antioxidants to human cells		Natural antioxidants (such as procyanidins), omega-3 fatty acids, VE, VC, and beta carotene, VP, selenium, and dietary fiber in addition to antioxidant-rich food, including tomatoes, garlic, nuts, oats, blueberries, broccoli, salmon, wheat, and green tea

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
