# Peer review of "Long-Term Space Nutrition: A Scoping Review"

_nutrients, 2021, doi:10.3390/nu14010194_

Round 1
Reviewer 1 Report
The authors provide a rich overview of the literature on nutrition in space. The scoping review is a very interesting read and provides a lot of food for thought on the challenges of advancing human habitat into space. The authors nicely describe the state of the art on the challenges to space nutrition and on possible solutions.
Main comments
- The paper is easy to read. However, there is some repetition within and across several sections. My suggestion would be to consolidate the information provided and reduce redundancy.
- Although several arguments are provided, the limitations of functional food for (longer-term) space missions are not entirely clear. It might help to provide an overview of/mention the need for space food systems earlier in the paper. Why are fresh fruits/vegetables needed in space and why they cannot be replaced with supplements?
Detailed comments
- Line 50: 135.8 meters or square meters?
- Line 71/72: Why the nutrient loss in space foods, why can this not be supplemented through functional food, but only through fresh foods?
- Line 79: Reference missing [cite]
- Line 508-510: Repetition of “animals have lower potassium (and potassium salts) than plants”
- Table 4: Please check last row of issue 1) for formatting and content
- Line 667/668: Meaning not clear of the sentence “If the food is lost during processing and storage, the crew will not get enough nutrition even if the food does provide sufficient nutrition”.
- Line 696-698: Repetition of the sentence and meaning not clear of “nutrient loss over 3 to 5 years is expected to be significant and may lead to undernutrition of the food system”. What is meant by undernutrition of the food system?
- Line 735: Double colon after “dissolution in packaging foods”
- Line 774: Superfluous “f” after “preserved for a long time,…”
Author Response
Point 1: Line 50: 135.8 meters or square meters?
Response 1: Line 67: 135.8 square meters
Point 2: Line 71/72: Why the nutrient loss in space foods, why can this not be supplemented through functional food, but only through fresh foods?
Response 2: Line 173: The paragraph before this sentence was largely rewritten to more clearly outline the deficiencies in the existing space nutrition system that largely relies on processed to better argue for the need to use fresh food materials.
Point 3: Line 79: Reference missing [cite]
Response 3: Line 128: The entire sentence has the same citation, which was moved from just before the comma to before the period for the sentence.
Point 4: Line 508-510 Repetition of “animals have lower potassium (and potassium salts) than plants”
Response 4: Line 1223: (and potassium salts) was removed
Point 5: Table 4: Please check last row of issue 1) for formatting and content
Response 5: Formatting issue was corrected
Point 6: Line 667/668: Meaning not clear of the sentence “If the food is lost during processing and storage, the crew will not get enough nutrition even if the food does provide sufficient nutrition”.
Response 6: Line 1286 (Line 7 of section 4.5): The last sentence of the first paragraph under section 4.5 was revised to provide more clarification.
Point 7: Line 696-698: Repetition of the sentence and meaning not clear of “nutrient loss over 3 to 5 years is expected to be significant and may lead to undernutrition of the food system”. What is meant by undernutrition of the food system?
Response 7: Line 1325: The last sentence of the 4th paragraph under section 4.5 was rewritten to provide more clarification.
Point 8: Line 735: Double colon after “dissolution in packaging foods”
Response 8: Line 1370 (Line 4 of the 3rd paragraph under section 4.6.1): The second colon was removed.
Point 9: Line 774: Superfluous “f” after “preserved for a long time,…”
Response 9: Line 1409 (Line 5 of Section 4.7.1.): “f” was changed to “the”

Reviewer 2 Report
This scoping review is relevant to understand the gaps and current evidence in the field of long-term space nutrition. It was reported that the review was conducted following the PRISMA-ScR checklist however a thorough study on the paper showed that some checklist of the methodology were not comprehensively reported. The authors should describe these:
i. state the process for selecting sources of evidence (that is, screening and eligibility) included in the scoping review.
ii. Describe the methods of charting data from the included sources of evidence and any processes for obtaining and confirming data from investigators.
iii. List and define all variables for which data were sought and any assumptions and simplifications made.
iv. If done, provide a rationale for conducting a critical appraisal of included sources of evidence; describe the methods used and how this information was used in any data synthesis.
v. Describe the methods of handling and summarizing the data that were charted.
vi. Report the results of the literature search, numbers of citations screened, duplicates removed and full documents (including the rationale for selection), ideally using PRISMA flow diagram template.
Generally the results and the discussion were well written however I recommend the authors provide three bullet point summary of their main findings and their implications.
Author Response
Point 1: state the process for selecting sources of evidence (that is, screening and eligibility) included in the scoping review.
Response 1: Line 620 (Section 2.2): The process for selecting sources for review and the screening/eligibility criteria are outlines in Section 2.2.
Point 2: Describe the methods of charting data from the included sources of evidence and any processes for obtaining and confirming data from investigators.
Response 2: Line 602 (Line 3 of Section 2): This is not a metadata analysis manuscript. As a result, some of the checklist items are not applicable. No data was obtained from investigators for charting.
Point 3: List and define all variables for which data were sought and any assumptions and simplifications made.
Response 3: Line 594 (Line 3 of Section 2: Materials and Methods): Some of the checklist items, such as the ones under Point 3, were not used because they were for meta-analyses, which are outside of the scope of this review.
Point 4: If done, provide a rationale for conducting a critical appraisal of included sources of evidence; describe the methods used and how this information was used in any data synthesis.
Response 4: Line 592 (Line 13 of Section 1.2: Rational and Objectives) The rational and objectives used for synthesizing the articles screened were summarized in Section 1.2
Point 5: Describe the methods of handling and summarizing the data that were charted.
Response 5: Line 602 (Line 3 of Section 2): This is not a metadata analysis manuscript. As a result, some of the checklist items are not applicable. No data was obtained from investigators for charting.
Point 6: Report the results of the literature search, numbers of citations screened, duplicates removed and full documents (including the rationale for selection), ideally using PRISMA flow diagram template.
Response 6: Line 605 (Line 4 of Section 2.1) Figure 1 illustrates the PRISMA review protocol used by two independent reviewers between January, 2020 and September, 2021.
Point 7: Generally the results and the discussion were well written however I recommend the authors provide three bullet point summary of their main findings and their implications.
Response 7: (Section 5): The first. second, and third major findings and implications were described in the first, second, and third paragraph of Section 5.

Round 2
Reviewer 2 Report
I want to commend the authors for improving their manuscript based on the recommendations- please accept in this present form for publication.